# Genetic Characterization of Raspberry Bushy Dwarf Virus Isolated from Red Raspberry in Kazakhstan

**DOI:** 10.3390/v15040975

**Published:** 2023-04-16

**Authors:** Mariya Kolchenko, Anastasiya Kapytina, Nazym Kerimbek, Alexandr Pozharskiy, Gulnaz Nizamdinova, Marina Khusnitdinova, Aisha Taskuzhina, Dilyara Gritsenko

**Affiliations:** 1Laboratory of Molecular Biology, Institute of Plant Biology and Biotechnology, Almaty 050040, Kazakhstan; m.kolchenko2020@gmail.com (M.K.); anastasiya.kapytina@mail.ru (A.K.); nazym_kerimbek@mail.ru (N.K.);; 2Department of Molecular Biology and Genetics, Al-Farabi Kazakh National University, Almaty 050040, Kazakhstan

**Keywords:** raspberry bushy dwarf virus, RBDV, RNA2, coat protein, movement protein

## Abstract

Raspberry bushy dwarf virus (RBDV) is an economically significant pathogen of raspberry and grapevine, and it has also been found in cherry. Most of the currently available RBDV sequences are from European raspberry isolates. This study aimed to sequence genomic RNA2 of both cultivated and wild raspberry in Kazakhstan and compare them to investigate their genetic diversity and phylogenetic relationships, as well as to predict their protein structure. Phylogenetic and population diversity analyses were performed on all available RBDV RNA2, MP and CP sequences. Nine of the isolates investigated in this study formed a new, well-supported clade, while the wild isolates clustered with the European isolates. Predicted protein structure analysis revealed two regions that differed between α- and β-structures among the isolates. For the first time, the genetic composition of Kazakhstani raspberry viruses has been characterized.

## 1. Introduction

Raspberry bushy dwarf virus (RBDV) is a representative member of the species *Idaeovirus rubi* of the genus *Idaeovirus* in the family *Mayoviridae*. *Idaeovirus rubi* is one of two species in the genus *Idaeovirus* [1]. RBDV naturally infects members of the *Rubus* species, including black raspberry (*R. occidentalis*), blackberry (*R. fruticosus*), loganberry (*R. loganobaccus*), boysenberry *(R. ursinus × idaeus*), and arctic bramble (*R. arcticus*) [2,3,4,5], as well as grapevine [6] and cherry [7]. It is transmitted both horizontally and vertically through pollen [8,9].

The RBDV genome is composed of two single-strand RNA molecules with sizes of 5.5 kb (RNA1) and 2.2 kb (RNA2), as well as a subgenomic 946 nt RNA molecule (RNA3) [10,11,12]. The bipartite RNA genome includes a bicistronic RNA2, while RNA3 is monocistronic [13]. RNA1 encodes a putative polymerase protein, while RNA2 encodes two genes, a 39 kDa putative movement protein (MP), whose nucleotide sequence is similar to movement proteins of other RNA viruses, and a 30 kDa coat protein (CP) [11]. The CP is expressed by RNA3, and it has been reported to play an important role in the infection process [14].

Several RBDV isolates have been distinguished based on the molecular and serological characteristics similar to Scottish-type D200 (S), resistance-breaking isolates (RB), and serological variant isolates from black raspberry (B) [15,16]. RBDV has been found to form virus complexes with raspberry leaf mottle virus (RLMV), increasing their infection synergy [17,18].

The genes located on the RNA2 segment are critical for transmission. The CP determines virion formation and its structure [19], provides assistance in replication via genome activation [14,20], and interacts with the plant immune system in tandem with the MP [21,22]. Despite significant genetic variability associated with CP and MP sequences, secondary and tertiary structures remain relatively conserved [23], indicating a morphological separation into a number of individual architectural classes [19].

This work aimed to identify and compare complete RNA2 sequences of RBDVs infecting both cultivated and wild subspecies of red raspberry from Kazakhstan and other countries around the world to gain insight into its evolution and epidemiology.

## 2. Materials and Methods

### 2.1. Sample Collection and Detection by RT-PCR

A total of 187 samples exhibiting leaf chlorosis (Figure 1) from a cultivated environment (Almaty Pomological Garden) and 35 samples from natural environments (Trans-Ile Alatau mountain range) were collected and analysed for RBDV infection. RNA was isolated from leaf tissue using the cetriltrimethylammonium bromide (CTAB) protocol [24]. The quality of the isolated RNA was confirmed by agarose gel electrophoresis (2% *w*/*v*). Reverse transcription was conducted using RevertAid Reverse Transcriptase (Thermo Scientific, Waltham, MA, USA). The mix of 200 ng RNA, 0.5 μM Oligo-dT, and 0.5 μM random hexamer primers in a final volume of 15 μL was incubated for 10 min at 72 °C and then cooled on ice. Then, 5× RT reaction buffer, 0.5 mM dNTPs, and 200 U reverse transcriptase were added, followed by incubation for 1 h at 45 °C [25].

The raspberry samples were tested for the presence of RBDV, as well as raspberry ringspot virus (RpRSV), raspberry leaf mottle virus (RLMV), and raspberry leaf blotch virus (RLBV) via PCR assays with specific primers targeting the conserved regions inside different protein genes (Table 1).

The primer design process occurred as follows. The known nucleotide sequences of viral isolates were retrieved from GenBank, then aligned by Muscle [26], ClustalW [27]. The specificity of the primers was tested via sequencing of the PCR product and subsequent NCBI-Blast and Primer-BLAST searches. The most suitable primer pairs were selected for detection. Two microlitres of cDNA were used in the PCR reaction mix, along with 5 U Taq DNA Polymerase (New England Biolabs), 0.5 μM of forward and reverse primers, and 0.5 mM dNTPs. The amplification programme consisted of 95 °C for 3 min, 30 cycles of 95 °C for 30 s, 52 °C for 20 s, and 72 °C for 40 s, followed by a final extension at 72 °C for 5 min [25].

### 2.2. Amplification of Genomic RNA2 by RT-PCR

The primer pair for amplification of RNA2 was designed to correspond to the 5′ and 3′ ends of RBDV RNA2 (Table 1), and was synthesised by Macrogen Inc. (Seoul, Republic of Korea).

The cDNA was synthesised using 250 ng of RNA template, 1.0 μM reverse primer, and 200 U SuperScript IV Reverse Transcriptase (Invitrogen). The amplification utilised Q5 High-Fidelity DNA polymerase (New England Biotechnology), Q5 Buffer, and Q5 Enhancer for high fidelity and was performed on the platform C1000 Touch^TM^ Thermal Cycler (Bio-Rad Laboratories, Hercules, CA, USA) under the following conditions: 96 °C for 3 min, 30 cycles of 96 °C for 30 s, 52 °C for 20 s, and 72 °C for 3 min, followed by a final extension at 72 °C for 10 min.

The yield was confirmed using gel electrophoresis, and the PCR product was subsequently purified using the GeneJet PCR Purification Kit (Thermo Scientific).

### 2.3. Nanopore Sequencing and Assembly

Library preparation was performed according to the “Ligation sequencing influenza whole genome (SQK-LSK109 with EXP-NBD196)” protocol from the Nanopore Community tab, with the substitution of the EXP-NBD196 kit for EXP-NBD104 for 12 samples. Barcode-labelled cDNA libraries were sequenced on a FLO-MIN106D flow cell using MinION (Oxford Nanopore Technologies, Oxford, UK). All sequences generated in this study were deposited into the NCBI GenBank database. Raw sequence reads were filtered based on quality score and read length using FastQC [28] and the MultiQC tool [29]. Epi2Me Labs (ONT) analysis confirmed the presence of RBDV.

Assembly was performed following the bioinformatics pipeline introduced by Brancaccio et al. for human papillomavirus [30]. After base calling the raw data on a “high” setting with Guppy [31], the results were filtered for reads between 500 and 2300 nucleotides long using Filtlong v.0.2.1 [32]. The worst 10% of the reads were discarded. The genome was assembled using Canu 2.2 [33] and polished with Medaka v1.7.2 [34] against filtered reads. To assess the effectiveness of the pipeline, contigs offered by Canu were analysed for open reading frames (ORFs) of 1077 and 825 nucleotides corresponding to sequences coding for MPs and CPs using UGENE v45.1 software [35] and translated and matched to the NCBI protein database using the BLASTp program (URL: http://blast.ncbi.nlm.nih.gov/Blast.cgi (accessed on 12 January 2023)). Positive matches were selected and trimmed down to the size of the reference sequence (NC_003740).

Whole genome sequences of the 18 RBDV isolates were submitted to GenBank (OQ336272–OQ336289). These sequences were analysed alongside publicly available sequences.

### 2.4. Molecular Characterisation and Phylogeny

The viral sequences were subsequently aligned with RNA2 sequences from the NCBI database using the MAFFT algorithm [36] implemented using UGENE v45.1 software [35]. The most genetically variable regions (deletions, insertions, or more than 2 substitutions across all samples) were plotted separately. A sequence logo for visualisation was generated in WebLogo [37].

Single nucleotide polymorphisms (SNPs) and insertion-deletion mutations (indels) called in BCFtools [38,39] were filtered by frequency, and only instances appearing in more than 30 reads were counted.

For each sample, mutations were defined as mismatches against the consensus sequence. Mutational frequency was calculated as the total number of unique mutations divided by the total number of nucleotides within the sequenced genomes [40,41], while mutational frequency per codon was estimated by summing the mutation frequencies of each codon within each gene, divided by the total number of codons within the sequenced genomes [42]. To arrive at mutant spectra heterogeneity, the normalised Shannon entropy was calculated according to the formula {$\Sigma i\;\left(p_{i}\times lnp_{i}\right)/lnN\}$, where *p_i_* is the frequency of each sequence in the mutant spectrum and *N* is the total number of sequences compared [40,41].

Phylogenetic trees were constructed using the maximum likelihood method in MEGA11 (Kimura-2 parameter model, bootstrap 100), RAxML (bootstrap 1000), and MrBayes (1 million iterations) and investigated for conserved clades [43,44,45,46,47,48].

## 3. Results and Discussion

### 3.1. Occurrence of RBDV in Kazakhstan

In 2020 and 2021, field surveys were conducted to observe the incidence of viruses in raspberry growing areas in Almaty province (ca. 700–750 m above mean sea level) as well as from high-altitude regions of the Tien Shan mountains, approximately 1100–1200 m amsl. A total of 222 leaf samples, including 187 raspberry plants from cultivated fields and 35 from wild growing areas, were tested by RT-PCR for the presence of RBDV, RLMV, RLBV, and RpRSV. RBDV was found in 47 cultivated and 15 wild plants, with an RLMV coinfection in nine and two samples, respectively. The two wild raspberry bushes infected with RLMV also harboured an RLBV infection. RpRSV was not detected in any of the samples. The coinfection of RBDV and RLMV will not be examined further in this study, as it is the topic of future research.

RBDV RNA2 was characterized in this study because the corresponding sequences from many isolates are available in GenBank [30], providing a better scaffolding for an informative phylogenetic tree to analyse relationships between RBDV isolates from different locations.

The samples which tested positive for RBDV were characterized by RT-PCR with a second set of primers covering the 5′ and 3′ untranslated ends of RBDV RNA2 (Table 1). Since this set of primers did not target a conserved region, an amplicon corresponding to the nearly full-length RNA2 sequence of only 18 isolates was obtained and used for Nanopore sequencing (Table 2). Two out of eighteen RBDV isolates sequenced (KZD8 and KZSelection4) were from plants with an RLMV coinfection.

### 3.2. Variability of Genomic RNA2

Pairwise comparison of RBDV RNA2 sequences revealed a high level of similarity (97–100%) among the Kazakhstani isolates, along with isolate R15 from the UK and DMSZ PV 1316 from the Netherlands. The latter did not find confirmation through the cladogram.

Phylogenetic analysis of complete RNA2 sequences was performed on those obtained in the present study and those available in GenBank (Table 2).

For the most part, RBDV isolates grouped based on the nature of the plant host. Cherry isolates from Turkey were closer to the grapevine isolates [7] (Figure 2). Separation of Kazakhstani isolates into three clusters was present in all three phylogenetic trees constructed using Bayesian (MrBayes, 1 million generations), Maximum Likelihood (RaxML, bootstrap 1000), and Neighbour-Joining (MEGA11, bootstrap 1000) methods (Figure 2, others not pictured) and suggests three separate introduction events.

Swedish isolate SE3 from wild raspberry in Uppsala [51] clustered with Kazakhstani isolates KZWild2, KZWild4, and KZSelection4 (crossbreed) from wild raspberry but were separate from RBDV isolates from cultivated raspberry. This could indicate that the immune systems of the wild plants evolved within the context of plant virus ecology, which is categorically different from managed crop varieties [53,54]. Evidence suggests that non-cultivated plants can succumb to a wide diversity of plant viruses [55], although some coexist on the principle of tolerance [56] and even mutual benefit [57]. The combination of such circumstances might cause similar traits in cultivated hosts, even in different geographical locations.

The second cluster consisting of cultivated isolates KZ3-4, KZMol3, KZMol6, KZMol11, KZSelection2-8, and KZOgonek was consistent across the three trees, and was located next to the sequences of crops from Belarus and the UK, suggesting that RBDV might have been introduced through the importation of infected planting material from Russia [51] or England. The third cluster of Kazakhstani RBDV isolates was situated at a distance from the available isolates, the majority of which originated in Europe. It is possible that, as more Asian isolates are sequenced, new phylogenetic relationships will be revealed.

Among the Kazakhstani samples, the mutation frequencies ranged between 0.95 × 10^−3^ and 1.19 × 10^−3^ (Table 3), remaining relatively uniform across individual plants from multiple locations. Within each gene, the degree of mutation was diverse, indicating recurring substitutions, while the number of indels remained low. The highest mutational frequency of 4.04 × 10^−3^ was found in the CP gene of KZ3-4; however, the mean complexity was higher for the MP region (3.39 × 10^−3^). Both CP and MP genes are subject to significant selection pressure [23], and the data suggest that their mutation rates are similar. Low and uniform Shannon entropy indicates similarity among viral quasispecies.

Within the RBDV RNA2 metagenome, 182 variable positions (defined as containing either more than three substitutions across the sample pool or any indel mutations) were detected (Figure 3). Of these, 85 were located within the MP gene (7.89% sites), and 70 were within the CP gene (8.48% sites). The 5′ and 3′ untranslated regions of RNA2 could not be analysed sufficiently due to many sequences from GenBank lacking them. The most prominent insertions and deletions occurred outside the protein coding regions.

To assess the impact of genome variability on the protein sequence, phylogenetic analyses were run on each of the proteins separately.

### 3.3. Protein Sequence Analysis

The MP gene sequences (Figure 4) demonstrated similar distributions. The groups remained consistent with the whole RNA2 tree and point to a shared protein ancestor of the wild Kazakhstani cluster (1) and wild SE3.

The CP gene amino acid sequence analysis (Figure 5) included additional 46 sequences currently available in GenBank for a total of 94 (Appendix A). Among the additions were the Turkish blackberry isolates, which introduced a new, well-supported clade to the phylogenetic tree in relation to RBDV-China (Figure 4). Grouping according to the host plant remained, but the number and distribution of subgroups changed. The cluster of RBDV isolates from wild raspberry (1) previously located in the vicinity of SE3 was now near Slovenian RR3 and RR5, and the formerly isolated cluster (3) was placed deep within the upper raspberry subclade as a direct descendant of the wild SE3. Sweden is unlikely to be the progenitor of the CP gene or the main centre of *Rubus* biodiversity [58]. Sequencing isolates from the centre of *Rubus* biodiversity, such as Yunnan, might reveal more conclusive results.

Amino acids 1–29 and 204–274 were identical in all isolates except those from blackberry, even though McFarlane et al. confirmed in mutagenic studies that neither the 226 C-terminal nor the first 15 N-terminal amino acids are essential for biological activity of the virus [14].

Inconsistencies between CP and MP cladograms (Figure 4 and Figure 5) suggest that despite both proteins being involved in viral intracellular movement during the infection process [23], they face different selection pressures from the plant’s immune system.

### 3.4. Secondary Structure Prediction

Secondary structure properties prediction software RaptorX v1.01 [59] revealed that while CP sequences were relatively conserved among local isolates, the MP had two variable domains around the 170th and 250th amino acids, both consisting of 15 aa (Figure 6). Comparison with other isolates indicated that the α-helical domain at 170 aa was present in samples from the UK, Belarus, and Kazakhstan (15 out of 18). Other analysed sequences, along with the cluster of RBDV isolates from wild raspberry (1), exhibited putative β-strand structures at the same position.

The domain around 250 aa remained consistently α-helical across the sequences from GenBank, while MP sequences within cluster (3) had two substitutions in the region, Ala254Thr and Pro257Ser, which altered the predicted secondary structure to a β-strand in place of an α-helix and would have wider implications for the 3D conformation. The protein topology detection tool [60] predicted no transmembrane domains, concluding that all residues were facing inside. The Enthalpy prediction tool [61] assigned the two hydrophobic regions with the lowest ΔG_app_ (5.59–5.73 and 7.61–7.75, respectively), although they were not negative enough to be classified as trans-membrane domains. These regions can potentially be the sites through which MP associates with the membranes instead [62].

There is a lack of information currently available on the topology and aetiology of *Idaeovirus* MPs in general and RBDV in particular. However, groups of researchers consider them to be related to the *Bromoviridae* family (particularly to alfalfa mosaic virus and tobacco string virus) [63,64,65,66,67]. The alfamovirus MP is known to be associated with plasmodesmata and accomplishes intracellular movement by increasing their size exclusion limit and forming tubuli [68,69].

The introduction of a new serine into the amino acid sequence could potentially strengthen the binding of the MP to the CP (if the AMV model of movement between cells was to be applied) due to the addition of another phosphorylation site [70]. C-terminal deletions in AMV have been found to affect tubule formation and the association between MP and CP, which are prerequisites for cell-to-cell and systemic movement [71], although to a lesser extent than N-terminal mutations.

Because all RNA molecules were extracted from plants showing visible signs of virus infection, this confirmation change did not affect the virulence capabilities of RBDV. Further in planta experiments would help elucidate the effect of the amino acid substitutions on the infection process.

## 4. Conclusions

In Kazakhstan, raspberry planting material is imported primarily from Europe and Russia, in most cases without testing for absence of viruses. Therefore, research on viruses and virus transmission is crucial to raspberry industry.

Currently, our understanding of the complexities of plant virus dynamics across agroecological boundaries is severely limited due to the lack of information on wild populations [72,73]. For the first time, raspberry viruses present in Kazakhstan have been investigated for phylogenetic relationships. By providing RNA sequences from both sides of the agricultural–wild interface from a previously underrepresented single region, we contributed to a more complete picture of RBDV diversity and distribution, as well as tracking its movement between cultivated and non-cultivated plant host communities.

## Figures and Tables

**Figure 1 viruses-15-00975-f001:**
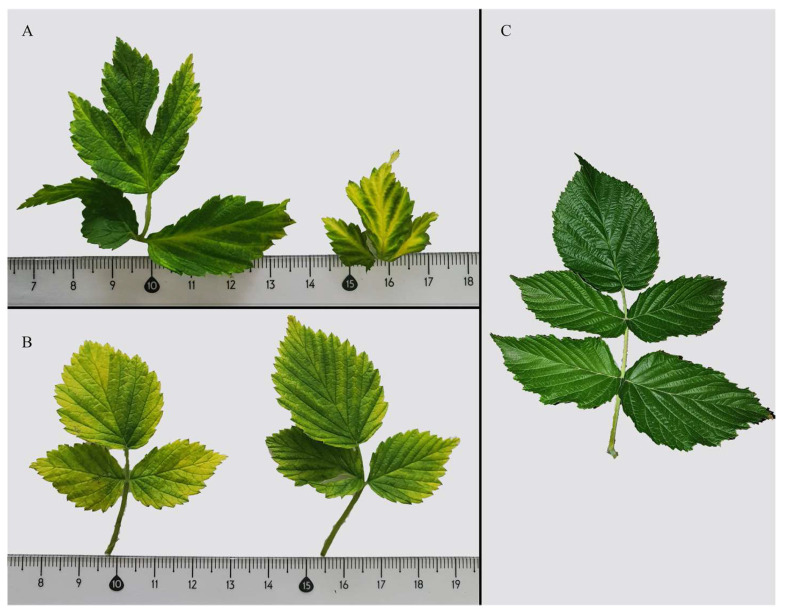
Plant images showing virus-like symptoms of RBDV infection in samples KZWild2 (**A**) and KZWild4 (**B**), as well as a healthy raspberry plant (**C**).

**Figure 2 viruses-15-00975-f002:**
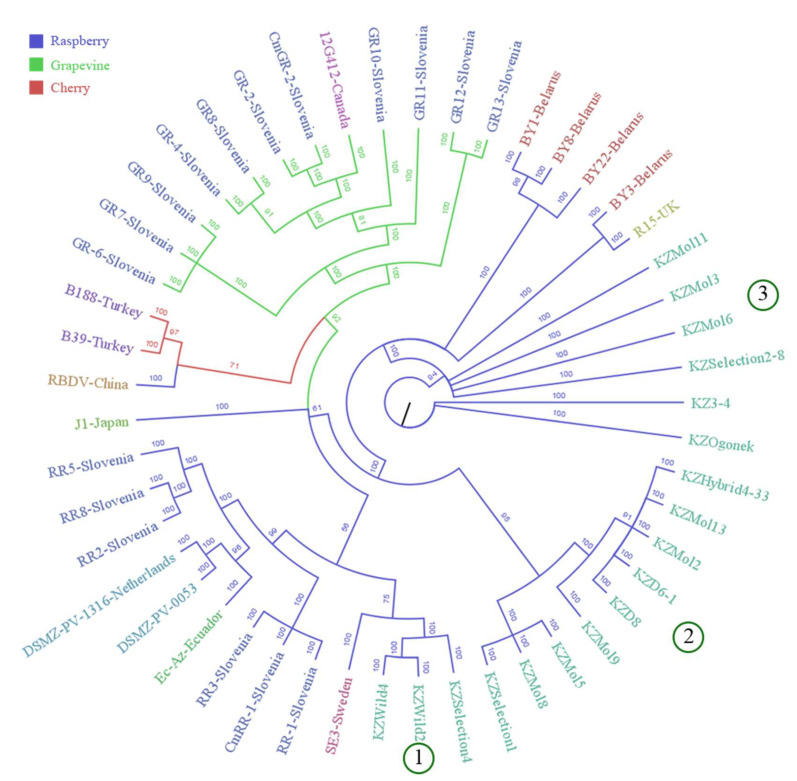
Cladogram constructed from all publicly available complete RNA2 sequences of RBDV using Bayesian algorithm. Branches are coloured according to the plant host, while isolates are coloured according to their countries of origin. Isolates introduced in this study (prefix KZ) are divided into three groups (1–3) according to their position in the cladogram.

**Figure 3 viruses-15-00975-f003:**
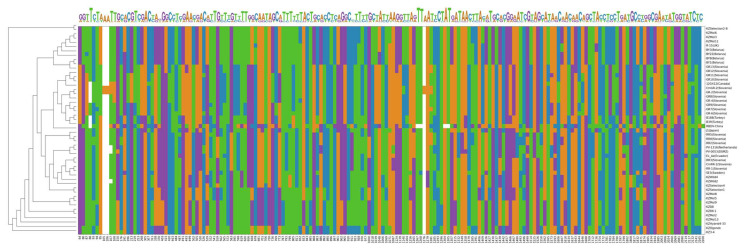
Heat map and logo of most variable regions within complete RBDV RNA2 sequences that were included in the phylogenetic tree.

**Figure 4 viruses-15-00975-f004:**
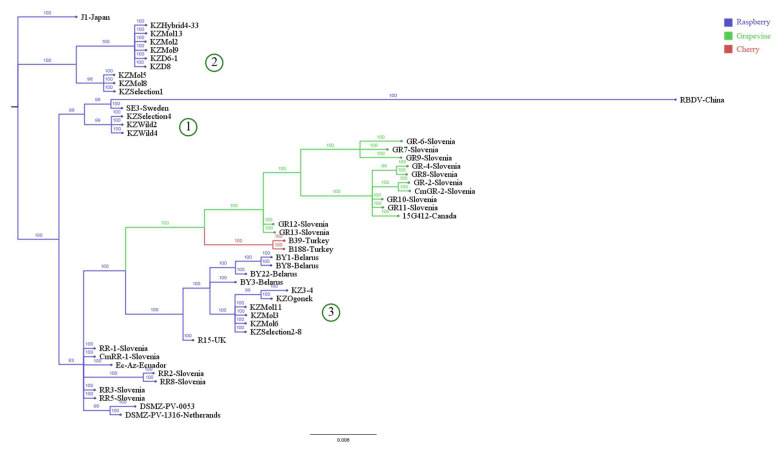
Phylogenetic analysis constructed from all publicly available MP sequences using Bayesian algorithm. Branches are coloured according to the plant host. Kazakhstani cluster (1), (2) and (3) are congruent with the Figure 3.

**Figure 5 viruses-15-00975-f005:**
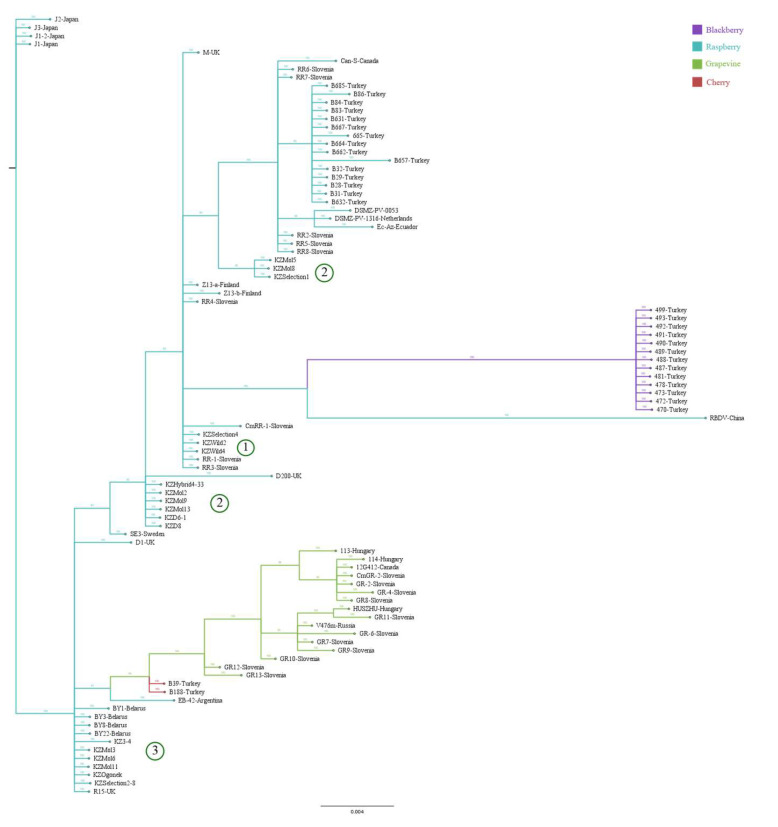
Phylogenetic analysis constructed from all publicly available CP sequences using Bayesian algorithm (includes additional 46 sequences). Branches are coloured according to the plant host. Kazakhstani cluster (1), (2) and (3) are congruent with the Figure 3. Both halves of the separated Kazakhstani cluster two are marked with (2).

**Figure 6 viruses-15-00975-f006:**
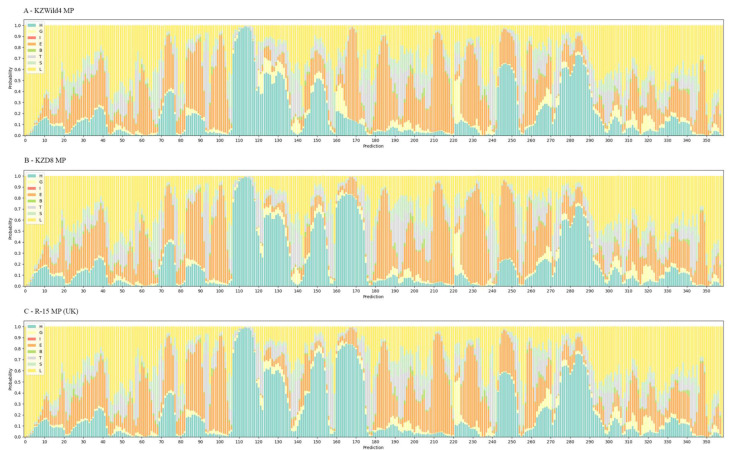
Protein Structure Property Prediction of KZWild4 MP, KZD8 MP and R-15 MP (reference) using SS8 DSSP notation. Note the prevalence of either E (extended strand, participating in β ladder) structures or H (α-helix) structures at 160–175 aa and 240–255 aa.

**Table 1 viruses-15-00975-t001:** Sequence information of primer pairs used in RT-PCR and PCR assays for detection and amplification of RBDV, RLMV, RLBV, and RpRSV [25]. NP = nucleocapsid protein.

Virus	Purpose	Region	Direction	Sequence (5′–3′)	Amplicon Size (bp)
RBDV	Detection	CP	F	agatccatgacggatgtgg	182
			R	aactaagttagaactattgtgg	
RBDV	Amplification	RNA2	F	agatccatgacggatgtgg	2231
			R	aactaagttagaactattgtgg	
RLMV	Detection	CP	F	tagcgtacttgtactgttc	163
			R	tacacttgtagcatgtttgg	
RLBV	Detection	NP	F	tacacttgtagcatgtttgg	106
			R	ccaacccttgtcaattttgat	
RpRSV	Detection	MP	F	cagagtatgggtgatttct	127
			R	gaaacagcgcactctt	

**Table 2 viruses-15-00975-t002:** Complete RBDV RNA2 sequences analysed in the present study.

Isolate	Accession	Country	Host	Year	Source
J1	AB948215	Japan	Red raspberry cv. Autumn Britten	2016	Direct submission
RBDV-China	DQ120126	China	*Rubus multibracteatus*	2003	[49]
GR-6	EU796085	Slovenia	Grapevine	2009	[50]
GR-4	EU796086	Slovenia	Grapevine	2009	[50]
GR-2	EU796087	Slovenia	Grapevine	2009	[50]
RR-1	EU796088	Slovenia	Red raspberry	2009	[50]
CmRR-1	EU796089	Slovenia	*C. murale*—raspberry	2009	[50]
CmGR-2	EU796090	Slovenia	*C. murale*—grapevine	2009	[50]
BY1	FR687354	Belarus	Red raspberry cv. Zolotye Cupola	2011	[51]
BY3	FR687355	Belarus	Red raspberry cv. Abricosovaya	2011	[51]
BY8	FR687356	Belarus	Red raspberry cv. Zolotye Cupola	2011	[51]
BY22	FR687357	Belarus	Red raspberry cv. Elegantnaya	2011	[51]
SE3	FR687358	Sweden	Red raspberry	2011	[51]
Ec_Az	KJ007640	Ecuador	*Rubus glaucus*—Andean rasp	2014	[16]
RR2	KY417868	Slovenia	Red raspberry cv. Chilliwack	2016	Direct submission
RR3	KY417869	Slovenia	Red raspberry	2016	Direct submission
RR5	KY417870	Slovenia	Red raspberry	2016	Direct submission
RR8	KY417871	Slovenia	Red raspberry cv. Titan	2016	Direct submission
GR7	KY417872	Slovenia	Grapevine cv. Chardonnay	2016	Direct submission
GR10	KY417873	Slovenia	Grapevine cv. Renski Rizling (Riesling)	2016	Direct submission
GR11	KY417874	Slovenia	Grapevine cv. Sipon	2016	Direct submission
GR12	KY417875	Slovenia	Grapevine cv. Zweigelt	2016	Direct submission
GR13	KY417876	Slovenia	Grapevine cv. Kraljevina	2016	Direct submission
GR8	KY417880	Slovenia	Grapevine cv. Modra Frankinja (Blaufrankisch)	2020	[52]
GR9	KY417881	Slovenia	Grapevine cv. Modra Frankinja (Blaufrankisch)	2020	[52]
12G412	MH802010	Canada	Grapevine	2019	Direct submission
PV-0053	MW582778	N/A	*Chenopodium quinoa* (lab) DSMZ PV-0053	2021	Direct submission
B39	MW729744	Turkey	Cherry	2022	[7]
B188	MW729744	Turkey	Cherry	2022	[7]
PV-1316	MZ202351	Netherlands	Red raspberry DSMZ PV-1316	2021	Direct submission
R15	S55890	UK	Red raspberry cv. Mailing Jewel	1991	[11]
KZ3-4	OQ336272	Kazakhstan	Red raspberry, crop	2021	Present study
KZD6-1	OQ336288	Kazakhstan	Red raspberry, crop	2021	Present study
KZD8	OQ336289	Kazakhstan	Red raspberry, crop	2021	Present study
KZHybrid4-33	OQ336273	Kazakhstan	Red raspberry, crop	2021	Present study
KZMol11	OQ336274	Kazakhstan	Red raspberry, crop	2021	Present study
KZMol13	OQ336275	Kazakhstan	Red raspberry, crop	2021	Present study
KZMol2	OQ336276	Kazakhstan	Red raspberry, crop	2021	Present study
KZMol3	OQ336277	Kazakhstan	Red raspberry, crop	2021	Present study
KZMol5	OQ336278	Kazakhstan	Red raspberry, crop	2021	Present study
KZMol6	OQ336279	Kazakhstan	Red raspberry, crop	2021	Present study
KZMol8	OQ336280	Kazakhstan	Red raspberry, crop	2021	Present study
KZMol9	OQ336281	Kazakhstan	Red raspberry, crop	2021	Present study
KZOgonek	OQ336282	Kazakhstan	Red raspberry, crop	2021	Present study
KZSelection1	OQ336283	Kazakhstan	Red raspberry, crop	2021	Present study
KZSelection2-8	OQ336284	Kazakhstan	Red raspberry, crop	2021	Present study
KZSelection4	OQ336285	Kazakhstan	Red raspberry, crop	2021	Present study
KZWild2	OQ336286	Kazakhstan	Red raspberry, wild	2021	Present study
KZWild4	OQ336287	Kazakhstan	Red raspberry, wild	2021	Present study

**Table 3 viruses-15-00975-t003:** Molecular characterization of mutant spectra for RNA2 of RBDV isolates: mutation frequency per genome and per gene. Numbers of SNPs, indels, nucleotide diversity, and Shannon index were estimated by retracting the 10^−3^ error rate correction.

		Mutation Frequency per Codon, 10^−3^				
Samples	Mutation Frequency, 10^−3^	MP	CP	SNP	Indel	Shannon Index
KZ3-4	1.19	4.02	4.04	30	0	0.00278
KZD6-1	0.97	3.25	2.42	21	0	0.00233
KZD8	0.97	3.25	2.42	21	0	0.00233
KZHybrid4-33	1.00	3.25	2.42	22	1	0.00238
KZMol11	0.97	3.25	2.42	21	1	0.00233
KZMol13	1.10	3.71	3.84	21	0	0.00258
KZMol2	0.95	2.94	3.43	21	0	0.00228
KZMol3	1.10	3.71	3.84	21	1	0.00258
KZMol5	0.95	2.94	3.43	21	0	0.00228
KZMol6	0.97	3.25	2.63	26	0	0.00233
KZMol8	1.10	3.71	3.84	20	0	0.00258
KZMol9	0.97	3.25	2.42	20	0	0.00233
KZOgonek	1.12	3.87	3.84	26	0	0.00263
KZSelection1	0.95	2.94	3.43	26	0	0.00228
KZSelection2-8	0.97	3.09	3.03	26	1	0.00233
KZSelection4	1.12	3.87	3.84	26	0	0.00263
KZWild2	1.10	3.40	3.23	25	0	0.00258
KZWild4	1.07	3.40	3.23	25	0	0.00253

## Data Availability

Raw data are available in the Appendix A.

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
