# Peer review of "Genetic Characterization of Raspberry Bushy Dwarf Virus Isolated from Red Raspberry in Kazakhstan"

_viruses, 2023, doi:10.3390/v15040975_

Round 1

Reviewer 1 Report

You had 222 samples of which 18 are positive for RBDV; this information is included in materials and methods and should be in results. The results section is jumping directly into comparing isolates from all over and it is hard to follow up on what happens with the isolates from Kazakhstan. Your objectives make the reader wanting to follow up on these isolates. One of the major issues is that you identified 18 samples positive by PCR and my understanding is no additional samples were sequenced; could it be possible there is greater diversity of RBDV and your primers did not work on some.

Also for the majority of plant viruses the replacase is the main target for evolutionary analysis. If you have performed nanopore sequencing you should have obtained both RNA1 and RNA2; why was this research only focusing on the RNA2?

  Line 25: Raspberry bushy dwarf virus is not member of the species  Idaeovirus rubi.

 Idaeovirus rubi is the scientific name for RBDV

Line 26:  Mayoviridae should be italicized

Line 42: raspberry does not need to be capitalized  

Line 59: replace according to with used

Lines 235: do not capitalize alfalfa and tobacco

Reviewer 2 Report

Manuscript viruses-2309351 describes the characterization of RNA2 of raspberry bushy dwarf virus (RBDV) from cultivated and wild raspberry in Kazakhstan and the analysis of RBDV RNA2 variability. The research is of interest but the manuscript is not well written. There are numerous language problems and the text is verbose. See editorial recommendations for improvement below.  Based on these limitations, manuscript viruses-2309351 is recommended for publication after revisions.

Specific comments:

Lines 2 and 261: Change Raspberry to raspberry

Line 3: ... from red raspberry in Kazakhstan

Linea 10-11: ... and grapevine that has also been found in Cherry. Most of the currently available RBDV sequences are from European raspberry isolates. this study ...

Line 11 and throughout the manuscript: Change RNA-2 to RNA2

Lines 11-12: ... sequence genomic RNA2 of RBDV isolates form both cultivated and wild raspberry in Kazakhstan and investigate ...

Line 13: ... and phylogenetic relationships. as well as ...

Lines 14-16: Eliminate these sentences

Line 18: Eliminate of interest

Line 20: Replace whole genome sequencing by RNA2

Lines 25-26: ... (RBDV) is a representative members of the species Idaevirus rubi of the genus Idaevirus in the family Mayoviridae ...

Line 31 and throughout the manuscript: Change RNA-1 to RNA1

Lines 32-33: ... as well as a sub-genomic RNA molecule of 946 nt ...

Line 33 and throughout the manuscript: Change RNA-3 to RNA3

Line 33: Is RBDV RNA1 really bicistronic? Please correct this sentence

Line 34: Eliminate in nature

Line 39: Several RBDV isolates have been ...

Line 40: ... characteristics similar to ...

Lines 41-43: This sentence could easily be eliminated without affecting the quality of the manuscript.

Line 48: Eliminate within each group

Line 56: A total of 187 samples from plants exhibiting leaf deformation and chorosis (Figure 1) from a cultivated ...

Lines 58-59: ... and analysed for RBDV infection. RNA was isolated ...

Line 71: Change capsid to coat for consistency

Lines 73-74: ... reveal conserved sites for primer design [27]. The specificity of primers was tested ...

Lines 82-83: this sentence belongs to the section on Results

Line 84: Amplification of RBDV genomic RNA2 by RT-PCR

Line 120: The viral sequences were subsequently aligned with RBDV RNA2 sequences ...

Line 145: ... from The Netherlands

Line 146: ... complete RNA2 sequences was performed on those obtained in this study and those available ...

Line 149, caption of figure 2: ... according to the plant host. Isolates ...

Line 149, caption of figure 2: Change 3 to three

Line 149, caption of figure 2: ... groups (1-3) according ...

Line 151: RBDV isolates grouped based on the nature of the plant host for the most part with Turkish cherry isolates more ...

Lines 152-153: Change samples to isolates

Lines 155-156: Eliminate supplementary figures S1 and S2. These two figures do not add any substantive information to the manuscript.

Line 156: ... and suggests three separate introduction events.

Lines 157-150: Swedish isolate SE3 from wild raspberry in Uppsala [50] clustered with Kazakhstani isolates KZWild1, KZWild4 and KZWild4 (crossbreed) from wild raspberry but separately from RBDV isolates from cultivars raspberry. this could indicated ...

Line 164: ... traits in cultivated hosts ...

Line 166: Eliminate it

Line 167-168: ... and the UK, suggesting that RBDV might have been introduced through the importation of infected planting material from Russia (50] or England. The third cluster of Kazakhstani RBDV isolates was separate from other isolates ...

Line 171: ... relations will be revealed.

Lines 175 and 184: Change InDels to indels, and InDel to indel

Lines 185 and 186: Eliminate coding

Line 186: The 5' and 3' untranslated regions of RNA2 could not be ...

Line 191, caption of figure 3: Heat map of most variable nucleotide regions within complete RBDV RNA2 sequences that were included in the phylogenetic tree

Line 193: the MO gene sequences ...

Lines 200-201: Eliminate within them

Line 201: Change the wild cluster to The cluster of RBDV isolates from wild raspberry

Figure 4, caption: ... Bayesian algorithm. Branches are colored according to the plant host.

Figure 4: Delineate the three sequence clusters of Kazakhstani RBDV isolates on the phylogenetic tree

Lines 204 and 205: Italicize Rubus

Lines 213-215: Eliminate this sentence

Figure 5, caption Change isolates to sequences

Figure 5: Delineate the three sequence clusters of Kazakhstani RBDV isolates on the phylogenetic tree

Line 245: ... visible signs of virus infection

Line 247: Italicize in plants

Line 247: ... experiments would help elucidate ...

Lines 247-248: Which phenomenon? Please clarify

Line 253: ... from a previously underrepresented single region, we contributed ...

Line 255: ... non-cultivated plant host communities.

Lines 255-259: Eliminate these sentences.

Table S1, legend: Change Raspberry to rasberry

Table S1, legend: Eliminate (CP)

Eliminate figure S1

Eliminate figure S2

Eliminate DSSP SS8 notation

Round 2

Reviewer 2 Report

Revisions have substantially improved manuscript viruses-2309351. Nonetheless, additional changes should be considered for further improvement. See recommendations below:

Line 16: Change RNA-2 to RNA2

Line 20: Add a concluding sentence

Line 26: … (RBDV) is a representative

Lines 70-71: The same raspberry samples were also tested for the presence of …

Line 72: … raspberry leaf blotch virus (RLBV) by RT-PCR.

Line 72: A reference or several references that describe RT-PCR conditions for the testing of RpRSV, RLMV and RLBV should be mentioned.

Line 77: … the coat protein gene. The known …

Line 158: Eliminate analysis

Line 160: … in nice and two samples, respectively.

Line 161: Change RpSV to RpRSV

Line 162: Change comorbidity to coinfection

Lines 164-165: RBDV RNA2 was characterized in this study because the corresponding sequence of many isolates is available in GenBank [31. A more …

Lines 167-169: Eliminate this sentence

Lines 170-177: Samples which tested positive for RBDV were characterized by RT-PCR with a second set of primers covering the 5’ and 3’ untranslated ends of RBDV RNA2 (Table 1). Since these primers did not target a conserved sequence region, an amplicon corresponding to the nearly full-length RNA2 sequence of only 18 isolates was obtained and used for Nanopore sequencing (Table 2). Two out of 18 RBDV isolates sequenced (KZD8 and KZSelection4) were from plants with an RLMV coinfection.

Line 190: Change cluster to isolates

Line 282: Change Alfamovirus to alfamovirus

Line 297: Eliminate dangerous

Line 298: … research on viruses and their transmission mode is …

Line 302: Change circulating to present

Line 328: Change is to are
